# Beyond the Black Box of Life Cycle Assessment in Wastewater Treatment Plants: Which Help from Bioassays?

**Michele Menghini** [1], **Roberta Pedrazzani** [1,2,*], **Donatella Feretti** [2,3], **Giovanna Mazzoleni** [4], **Nathalie Steimberg** [4], **Chiara Urani** [5], **Ilaria Zerbini** [3] and **Giorgio Bertanza** [6]

1   DIMI-Department of Mechanical and Industrial Engineering, University of Brescia, Via Branze 38, I-25123 Brescia, Italy
2   MISTRAL-Inter-University Research Center "Integrated Models for Prevention and Protection in Environmental and Occupational Health", DSCS, Department of Clinical and Experimental Sciences, University of Brescia, Viale Europa 11, I-25123 Brescia, Italy
3   DSCS-Department of Clinical and Experimental Sciences, University of Brescia, Viale Europa 11, I-25123 Brescia, Italy
4   DSMC-Department of Medical and Surgical Specialties, Radiological Sciences and Public Health, University of Brescia, Viale Europa 11, I-25123 Brescia, Italy
5   DISAT-Department of Earth and Environmental Sciences, University of Milan—Bicocca, Piazza della Scienza 1, I-20126 Milano, Italy
6   DICATAM-Department of Civil, Environmental, Architectural Engineering and Mathematics, University of Brescia, Via Branze 43, I-25123 Brescia, Italy
*   Correspondence: roberta.pedrazzani@unibs.it; Tel.: +39-030-3715505; Fax: +39-030-3702448

**Abstract:** The assessment of the environmental footprint of an organization or product is based on methods published by the European Union Joint Research Centre, which take 16 impact areas into account. Among the listed categories are human and freshwater ecosystem toxicities. Standard protocols utilize just chemical parameters as input data, hindering the determination of the full impact of complex mixes, such as pollutants released into the environment. Biological assays enable us to overcome this gap: in the present work, assays were employed to determine both baseline and specific toxicity to aquatic species (green algae, luminescent bacteria, and crustacean cladocera) as well as specific toxicity (mutagenicity and carcinogenicity). Ecological footprint was estimated with regard to the impact categories "freshwater toxicity" and "human cancer toxicity" following the standard methodology. In parallel, the impact on the above categories was estimated using the results of biological assays as input. Standard and bioassay-based results are not always congruent, and conventional methods generally underestimate the effects. Likewise, the choice of reference substance (metals or organics) influences the quantification of impact. Appropriate batteries of biological assays could therefore be utilized to complement LCA (Life Cycle Assessment) techniques in order to make them more sensitive when considering toxicity in mid-term impact categories.

**Keywords:** activated sludge; carcinogenic; ecotoxicity; effluent; environmental footprint; impact category; MBR; non-carcinogenic; toxicity

## 1. Introduction

Increasing awareness and sensitivity to the environmental effects of production and consumption patterns has highlighted the need to improve the sustainability of industries and consumers [1,2]. Life cycle assessment (LCA), based on ISO 14040 and ISO 14044, is a standardized tool that may help decision-makers develop a strategic plan based on environmental aspects [3–5]. This methodology quantifies the environmental impacts in all the stages of the process, from raw material withdrawal to the final disposal (i.e., from cradle to grave), to improve the environmental performance of products/organizations along their life cycle or compare different services/products in terms of sustainability [6–8]. Based on the ISO 14040 and ISO 14044 standards, several methodological approaches were

developed to carry out a life cycle impact assessment yielding a puzzling situation for both producers and consumers [9]. In 2011, the European Commission Joint Research Centre (EC-JRC) published the International Reference Life Cycle Data System (ILCD) Handbook recommendations to detail guidance and standards for applying LCA with quality and robustness. Hence, the Recommendation 2013/179/EU defined a uniform method for assessing and disclosing the environmental footprint of a product (PEF) and organization (OEF), developing a single market for green product initiatives. From 2013 to 2018, during a pilot phase, volunteering companies developed product and sector rules (PEFCR and OEFCR, respectively) which identified a category benchmark. In 2019, a subsequential transition phase began, with the extension of the category rules to other products/sectors and the adoption of policies implementing these procedures [9]. Finally, in 2021, the EU issued a more detailed recommendation (2279/2021/EU) for gathering the information during the pilot phases [7].

Alongside the fruitful applications in a variety of manufacturing sectors (from textiles to detergent products to food), in the field of wastewater treatment facilities, the PEF/OEF can be utilized to determine the optimal upgrade choices or the optimal operation decision from an environmental footprint perspective. Specifically, this methodology can be applied to estimate the overall environmental impact in terms of both direct impacts and indirect impacts [10–22]. The direct impacts are linked to effluents and other emissions, whereas the indirect impacts are linked to energy and resources consumed. The PEF/OEF allows for the establishment of an impact score for each of the sixteen categories listed in the Recommendation. It is based on the mass flows of pollutants released in different environmental contexts as well as the resources used. Since it is extremely difficult to quantify every pollutant (primary and secondary) present in an emission, non-targeted analyses have been receiving more attention, in an effort to describe these complex mixtures more accurately while going beyond the lists of standards outlined in regulations and guidelines [23–25]. The execution of bioassays has been increasingly proposed. They might be organized in batteries, including various modes of toxicity action, organisms with different biological complexities and trophic roles) [26–32]. The effects on living organisms exerted by exposure to wastewater, effluents, pre-potable water, and potable water can be estimated by biological assays. Instead of focusing on a subset of known pollutants, this last technique makes use of the biological reaction by combining both known and undiscovered substances. Several researchers have applied this bioanalytical approach to wastewater [30–34], drinking water [35–37], and surface water [38].

In this work, in addition to the standard technique, we have implemented an alternate PEF/OEF procedure based on the results of bioassays at three wastewater treatment plants that use conventional and advanced treatment processes.

The parallel application of the two procedures (current and innovative) was successfully carried out on the case studies selected on the account that the authors knew them very well, having subjected them to numerous and extensive monitoring; thus, reliable and historical data sets were available. Because of their unique sensitivity, the results suggest that bioassay performance might be incorporated into LCA techniques.

## 2. Materials and Methods

### 2.1. Wastewater Treatment Plants (WWTPs)

The three wastewater treatment plants, chosen as case studies, are located in Northern Italy.

WWTP A (design size 370,000 population equivalent, p.e.) is a conventional activated sludge system treating municipal wastewater with the contribution of agri-food industrial discharge (Figure 1a).

WWTP B (design size 60,000 p.e.) consists of a conventional activated sludge system. This plant has a significant winery effluent during the grape harvest period (September and October) that increases the organic pollution load. The scheme of the plant is depicted in Figure 1b.

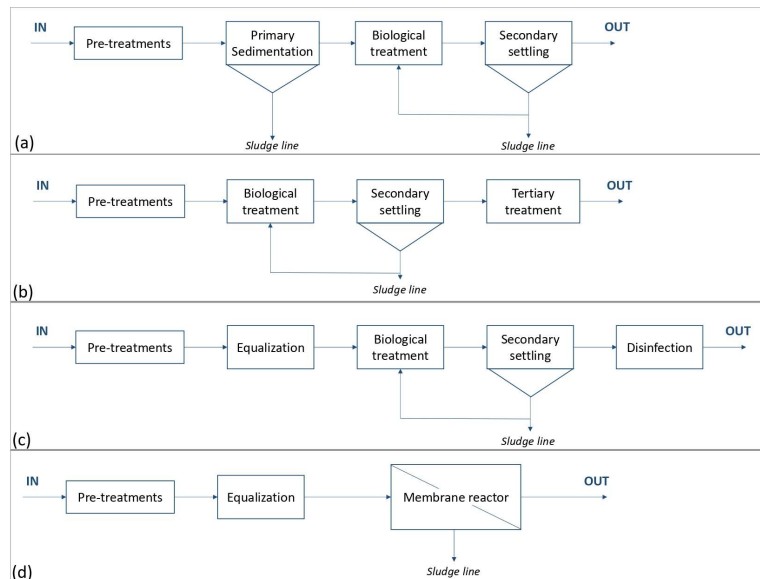

**Figure 1.** Process flow diagrams of the case studies: (**a**) WWTP A, (**b**) WWTP B, (**c**) WWTP C (CAS), (**d**) WWTP C (MBR).

WWTP C (design size 380,000 p.e.) has three process lines and primarily treats municipal wastewater: two CAS (conventional activated sludge) lines and one MBR (membrane bioreactor) line. The two treatment lines are depicted in Figure 1c,d, respectively.

A detailed description of the WWTPs and their main operational parameters are reported in [29,39].

### 2.2. Sampling Procedures

All the effluents were monitored using 24-h flow proportional composite samples (refrigerated autosampler). Each period of monitoring lasted two weeks. The samples were collected in WWTP A and C during a single monitoring campaign, whereas in WWTP B, two monitoring campaigns were conducted to account for both the routine time and grape harvest period. Ref. [40] returns more detailed information on the sampling procedure, while Ref. [41] summarizes sample pre-treatments after collection.

### 2.3. PEF/OEF Procedures

The goal of this research is a comparison between the environmental footprint obtained with the conventional method and the environmental impact calculated from the bioassay outcomes, using 1 m$^3$ of treated wastewater as a functional unit.

This study focuses on the direct emissions of a sewage treatment plant, namely the liquid emissions. These have a direct effect on the aquatic ecology and the water body from which water can be extracted for drinking/recreational uses, etc. In subsequent studies, direct gaseous and solid emissions (in this case, biological and chemical sludge), as well as any indirect emissions (raw material consumption, energy consumption, sludge formation, waste production), will be gradually incorporated. In this study, however, we chose to focus on liquid emissions because they account for the majority of flowrate and load. With respect to freshwater bodies and carcinogenicity, the effect categories evaluated concern both ecology and human health. The system's boundaries correspond to those of the case study's WWTPs.

During the inventory phase, primary data were collected in order to create an input data folder. Primary data include effluent characteristics (for the conventional environmental footprint method) and bioassay results (for the alternative environmental footprint approach). Effluent properties include the standard parameters (BOD, COD, TSS, N, P, and others), as well as metals, semimetals, and organic compounds (see Figure 2). The results of the chemical survey are reported and discussed in [29,34] because the subsets of

substances were quantified in a previous phase of the research, focused on evaluating the environmental impact in terms of released pollutants.

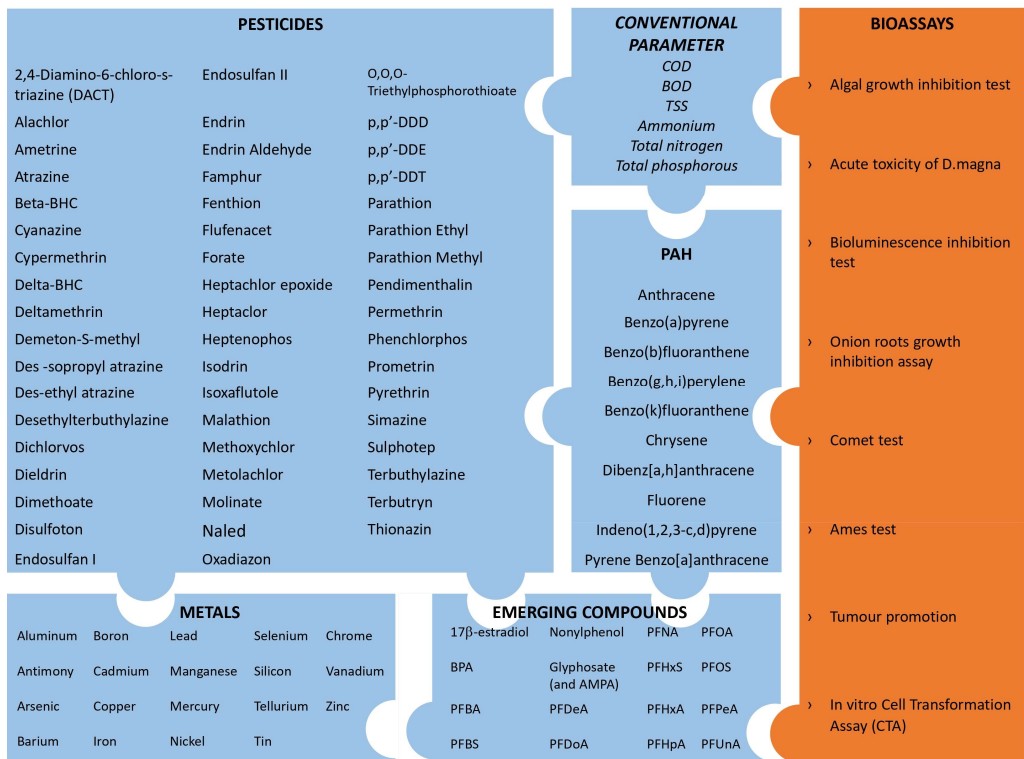

**Figure 2.** Subsets of compounds analyzed and bioassays used in the life cycle inventory assessment.

In addition to undergoing chemical testing, the same composite samples were put through a total of eight different types of bioassays. The goals of these tests were to study different modes of action and test species that lived at different levels of the trophic web. Four different bioassays were carried out in order to evaluate the cytotoxicity:

- algal growth inhibition test [42]
- crustacean cladocerans acute toxicity [43]
- inhibition of bacteria bioluminescence [44]
- inhibition of onion root growth [45]

Baseline toxicity identifies damaged living cells, which may result in planned or unplanned cell and tissue death [28]. Two different bioassays were utilized in order to detect genetic damage: the Comet test [46–48] and the Ames test [46,49].

The Comet assay detects damaged DNA in human leukocytes, whereas the Ames test highlights point reverse mutations in *S. typhimurium* strains (frameshift mutation, TA 98, and base-pair substitution, TA 100) with and without metabolic activation (S9). Finally, to assess carcinogenicity, two ecotoxicological assays were performed: tumor promotion and in vitro Cell Transformation Assay (CTA). The first test is related to the duplication of initiated cells (progression phase), whereas the second test is related to the subsequential phase in which unstable promoted cells become stable malignant tumors (progression phase) [28]. Ref. [41] provides detailed information.

The subsequent stage of the life cycle involves calculating the impact. Thus, the magnitude of potential environmental impacts of discharged water was calculated using the USEtox model for both human toxicity cancer and freshwater ecotoxicity, as shown in the Equation (1) below:

$$IS = \sum_i \sum_x CF_{x,i} \times m_{x,i} \tag{1}$$

where:

1. *IS*: Impact Score for the considered impact category
2. $CF_{x,i}$: Characterization Factor of the substance $x$ emitted to compartment $i$
3. $m_{x,i}$: the emitted $m$ ass of substance $x$ to compartment $i$

This model depicts the relative importance of each emission reported in the inventory. The calculation of impact category indicators in the PEF/OEF methodologies is based on mass flows of measured chemicals in discharged water. The environmental footprint, based solely on chemical analyses, serves as the standard against which the impacts obtained through the alternative procedure are measured.

The novel methodology used to calculate the environmental impact of these two categories is based on biological equivalent concentration. Using a dose–response calibration curve of a reference compound, the bioassay results were converted into biological equivalent concentrations. The biological equivalent concentration is the amount of the reference compound that produces the same effect as the tested mixture (i.e., discharge water). In this case, five reference substances (inorganics: cadmium and zinc; organic: 3,5-dichlorophenol, dodecylbenzene sulphonic acid, and maleic hydrazide acid) were used to create specific scenarios for each wastewater treatment plant. The remaining four ecotoxicological assay results were also used to evaluate the "human toxicity/cancer" category, where three reference substances (3-methylcholanthrene, lindane, and methyl methanesulfonate) yield specific scenarios for each wastewater treatment plant.

Two different datasets of characterization factors (CFs) were used in both environmental footprint methods to evaluate how CFs affect the impact score. The first CF dataset was obtained from the International Reference Life Cycle Data System (ILCD v.1.09), while the second was taken from the EF 3.0 package.

The United Nations Environment Program (UNEP)-Society for Environmental Toxicology and Chemistry (SETAC) Lyfe Cycle Initiative conducted an overall comparison of seven life cycle impact assessment toxicity characterization models (CalTOX, IMPACT 2002, USES-LCA, BETR, EDIP, WTSON, and EcoSense) in 2005 to develop a scientific consensus model. The USEtox 1.01, a meaningful toxic impact characterization model that calculates the CFs for freshwater ecotoxicity and human toxicity, was the result of this process [50]. As demonstrated in Equation (2), CFs are obtained by multiplying together three matrices that are each filled with the respective factors for the consecutive processes of fate (FF), exposure (XF), and effects (EF).

$$\overline{CF} = \overline{EF} \times \overline{XF} \times \overline{EF} \tag{2}$$

USEtox expresses CFs in two ways, depending on the impact category. For freshwater ecotoxicity, the unit is the potentially affected fraction of species (PAF) integrated over the freshwater volume (m$^3$) and the duration of 1 day (d) per kg of emission (PAF m$^3$·d/kg), whereas for human toxicity, the unit is the number of disease cases per kg of emission (cases/kg). The CFs are then summarized as comparative toxic units (CTU) per kg of emission in the software calculation (specifically CTUe/kg for the first examined category and CTUh/kg for the second). USEtox has been included in the ILCD recommendations and the EU Commission Product and Organization Environmental Footprint 2013/179/EU (PEF/OEF) in its pristine version 1.01. Following the emergence of several concerns during the PEF pilot phase's use of the USEtox 1.01, the PEF Technical Advisory Board (TAB) removed freshwater ecotoxicity, human cancer, and human non-cancer toxicity impact categories from the mandatory impact categories to be communicated in the context of PEF/OEF in December 2016. Using physicochemical and toxicity data from the Registration, Evaluation, Authorisation and Restriction of Chemicals (REACH), European Food Safety Authority (EFSA), and Pesticide Properties Database (PPDB) databases, the EC-JRC calculated new freshwater ecotoxicity characterization factors for 6011 substances, 3423 CFs for human toxicity/non-cancer, and 621 CFs for human toxicity/cancer using the USEtox 2.1. The three impact categories mentioned above are then recommended to be used in

the EF context with the level of recommendation III (recommended but to be employed with caution) [51].

Tables 1 and 2 show the values of the characterization factors of the adopted reference substances based on the ILCD and the EFs inventory, respectively, for freshwater ecotoxicity and human toxicity/non-cancer.

**Table 1.** Characterization factors for the freshwater ecotoxicity reference substances. The ILCD package represents old CFs, whereas the EF 3.0 package represents new CFs.

| Reference Substance | ILCD | EF 3.0 |
|---|---|---|
| Cadmium | 9710 | 229,000 |
| Zinc | 38,600 | 1330 |
| 3,5-dichlorophenol | 6910 | 50,780 |
| Dodecyl benzene sulfonic acid | 3110 | 11,963 |
| Maleic hydrazide | 182 | 2175.2 |

**Table 2.** Characterization factors for human toxicity/non-cancer reference substances. The ILCD package represents old CFs, whereas the EF 3.0 package represents new CFs.

| Reference Substance | ILCD | EF 3.0 |
|---|---|---|
| Methyl methanesulfonate (MMS) | $2.51 \times 10^{-6}$ | $2.51 \times 10^{-6}$ |
| Lindane | $3.35 \times 10^{-5}$ | $1.10 \times 10^{-4}$ |
| 3-methylcholanthrene (3MCA) | $4.82 \times 10^{-5}$ | $4.08 \times 10^{-5}$ |

## 3. Results and Discussion

### 3.1. Standard Approach: Open Issues Intrinsic to the Methodology

A thorough examination of the protocol (as in the European Recommendation 2279/2021/EU) reveals some flaws in the methodology that introduce uncertainty into the final impact score. The following are the shortcomings (●,♦,∗,~):

● The characterization factor's inherent variability.

The characterization factor is an important parameter in determining the impact score. It is influenced by the results of a battery of bioassays, as well as physicochemical and fate properties (such as n-Octanol/Water partition coefficient, water solubility, vapor pressure, and so on). Indeed, these values represent the USEtox model's input data, which generate the characterization factor. The physicochemical data are classified into three quality levels (low, intermediate, and high) based on the reliability, adequacy of the study, type of study, qualifier, and compliance with good laboratory practices. Only the data labeled as "high" in terms of the quality score are retained for each substance and each physicochemical parameter. When more data associated with the same quality score are available, the geometric mean is calculated to generate a unique value. However, not all the chemicals were submitted to the same number and type of bioassays (both in terms of lifetime and tested organism), resulting in toxicity data of varying trustworthiness.

As a result, new reliability input data and/or an update to the USEtox model can alter the value of the CFs and, as a result, the impact score. As shown in Tables 1 and 2, the updated version of USEtox, as well as some adjustments or additions to physicochemical and toxicity data, resulted in a significant change in the impact score.

The increased impact score in freshwater ecotoxicity (Table 3) is primarily due to aluminum, a substance for which no factors were anticipated in the ILCD factor set. This substance has a significant impact on the assessment's impact factor, accounting for between 89 and 98 percent.

**Table 3.** Impact on freshwater ecotoxicity category: values of CTUe/m$^3$ for the studied WWTPs.

| WWTP | ILCD CTUe/m$^3$ | EF 3.0 CTUe/m$^3$ |
|---|---|---|
| A | 4.27 | 73.62 |
| B (routine time) | 0.63 | 39.96 |
| B (grape harvest time) | 0.63 | 45.40 |
| C (CAS) | 8.94 | 35.25 |
| C (MBR) | 6.96 | 23.42 |

Unlike the previous category, the environmental footprint obtained with the new CF is significantly reduced in the human toxicity/cancer category because the CF of arsenic, nickel, mercury, and lead have been diminished by an order of magnitude (see Table 4).

**Table 4.** Impact on human toxicity category: values of CTUh/m$^3$ for the studied WWTPs.

| WWTP | ILCD CTUh/m$^3$ | EF 3.0 CTUh/m$^3$ |
|---|---|---|
| A | $1.73 \times 10^{-9}$ | $4.62 \times 10^{-10}$ |
| B (routine time) | $6.71 \times 10^{-10}$ | $1.22 \times 10^{-10}$ |
| B (grape harvest time) | $8.63 \times 10^{-10}$ | $1.73 \times 10^{-10}$ |
| C (CAS) | $2.26 \times 10^{-9}$ | $7.02 \times 10^{-10}$ |
| C (MBR) | $2.45 \times 10^{-9}$ | $7.61 \times 10^{-10}$ |

♦ The expression of the limit of quantification (LOQ) in the chemical analyses.

There are three possible approaches to consider the concentration of released substances detected below the LOQ:
(1) Nil
(2) Half the LOQ value
(3) Equal to the LOQ
The conventional procedure follows the hypothesis 1; however, a concentration below the LOQ does not automatically imply the absence of the substance. Hypotheses 2 and 3 represent a safe condition that raises the impact score. Indeed, as shown in Table 5, the three hypotheses result in impact score values that differ by up to two orders of magnitude.

**Table 5.** Environmental footprint on freshwater ecotoxicity category (expressed as CTU$_e$/m$^3$) and human toxicity/cancer category (expressed as CTU$_h$/m$^3$) calculated according to the standard approach detailed in Recommendation 2279/2021/EU.

| WWTP | Standard Approach | | | | | |
|---|---|---|---|---|---|---|
| | Freshwater Ecotoxicity | | | Human Toxicity/Cancer | | |
| | Hyp. 1 | Hyp. 2 | Hyp. 3 | Hyp. 1 | Hyp. 2 | Hyp. 3 |
| A | 73.62 | 919.89 | 1766.16 | $4.62 \times 10^{-10}$ | $6.56 \times 10^{-7}$ | $1.31 \times 10^{-6}$ |
| B (routine time) | 39.96 | 886.29 | 1732.62 | $1.22 \times 10^{-10}$ | $6.56 \times 10^{-7}$ | $1.31 \times 10^{-6}$ |
| B (grape harvest time) | 45.40 | 891.73 | 1738.06 | $1.73 \times 10^{-10}$ | $6.56 \times 10^{-7}$ | $1.31 \times 10^{-6}$ |
| C (CAS) | 35.25 | 255.75 | 476.25 | $7.02 \times 10^{-10}$ | $4.01 \times 10^{-9}$ | $7.32 \times 10^{-9}$ |
| C (MBR) | 23.42 | 243.92 | 464.42 | $7.61 \times 10^{-10}$ | $4.01 \times 10^{-9}$ | $7.32 \times 10^{-9}$ |

Because all the contaminants were below their quantification limits, hypotheses 2 and 3 theoretically would lead to a baseline impact score that can be applied to any WWTP. As a result, this factor introduces a new level of unpredictability to the impact score.

Finally, another critical issue is the possibility of different measurement methods used in different laboratories, resulting in different LOQs for the same substance. As a result, comparing the environmental footprint values becomes difficult.

∗    Regulated, known, and unknown chemicals.

The iceberg model proposed by [28] explains another level of uncertainty in the impact score calculated using the standard protocol. The wastewater monitoring program evaluates the regulated chemicals, which are a subset of the known substances, on a regular basis (the tip of the iceberg model). A significant unknown group of chemicals, representing the submerged portion of the iceberg, remains in the dark. In spite of the progress that has been made in analytical chemistry, it would be practically unfeasible to detect all the constituents of a mixture like wastewater. Even though suspect and non-target screening can identify far more chemicals than target analyses, unknown substances (such as metabolites, for instance) cannot be identified.

Furthermore, because this methodology focuses the evaluation on a subset of compounds for which a characterization factor has been defined, the assessment of the real environmental impact of WWTPs is severely limited. Indeed, taking into account the most recent version of the characterization factor (EF 3.0), some emerging compounds detected in the effluent above their limit of quantification (such as PFAS, per and polyfluoroalkyl substances, and AMPA, $\alpha$-amino-3-hydroxy-5-methyl-4-isoxazole-propionic acid) are not correlated with a characterization factor, effectively excluding substances from the evaluation. As a result, the environmental footprint calculated using this protocol does not provide a comprehensive assessment of the true impact of the WWTPs.

~    Mixture effect.

Chemicals exist as mixtures in wastewater effluent and in every environmental sample. When measuring the level of toxicity of the sample, one of the most important considerations to take into account is how the various substances in the combination interact with one another. In fact, even if a substance is detected below its LOQ, the effect of the mixture might be unsettling [28].

The traditional protocol only considers the additive effects. The environmental footprint is the sum of the impacts caused by the detected substances. This aspect adds another layer of uncertainty because the synergistic/antagonistic effect can increase/decrease the impact score.

### 3.2. Alternative Approach: A Step towards Overcoming Weak Points

The results of the bioassays are utilized in the novel strategy, which changes the focus from the toxicity of the individual compounds to the toxicity of the mixture as a whole. It addresses the following issues that were found in the conventional approach in an effort to increase the robustness of the impact score:

◆    The expression of the limit of quantification (LOQ) in the chemical analyses.

Instead of the emitted chemical concentration, the alternative approach uses the corresponding concentrations as input data. This correction eliminates the LOQ issue because each response in the biological test corresponds to a value of biological equivalent concentration.

∗    Regulated, known, and unknown chemicals.

~    Mixture effect.

Despite their inability to resolve single compounds, bioassays provide a comprehensive picture of the toxicity exerted by all chemicals present in a sample. The observed responses reflect a mixture of known and unknown substances (such as transformation products) measuring toxicity while considering the true interactions between the single components. As a result, the innovative procedure returns an impact score that considers the entire iceberg of pollutants rather than just the tip of the iceberg, as stated in the conventional approach. Chemicals can affect living organisms in a variety of ways. Some have non-specific effects, while others have specific toxic effects or reactive toxicity. As a result, it is critical to developing a battery of bioassays that includes both in vivo and in vitro assays.

It is worth noting that the novel method introduces a new level of variability in comparison to the traditional protocol. Indeed, the response of the tested organism is

characterized by inherent variability in terms of experiment repeatability. This is a new weakness introduced by the alternative methodology, which is added to the first criticality (●).

Overall, the alternative protocol reduces the total number of criticalities, lowering the impact score's variability.

### 3.2.1. Comparing the Protocols: Freshwater Ecotoxicity

Based on the various reference compounds and bioassays performed, 17 scenarios for each wastewater treatment plant were developed as part of the novel procedure (see Figures 3 and 4). The red dotted lines represent the impact scores calculated considering the substances whose concentrations were > their relative LOQs. A thorough examination of the impact scores derived by the alternative approach reveals that estimations based on metals as reference compounds result in lower impact scores for all plants and tests, except for the result of the luminescent bacteria assay developed with cadmium as the reference compound (only for WWTP A). In contrast, the estimation based on organic reference substances resulted in a high impact score. Furthermore, a detailed examination of the graphs reveals that the impact score based on the DCF as a reference substance has much more consistent effects.

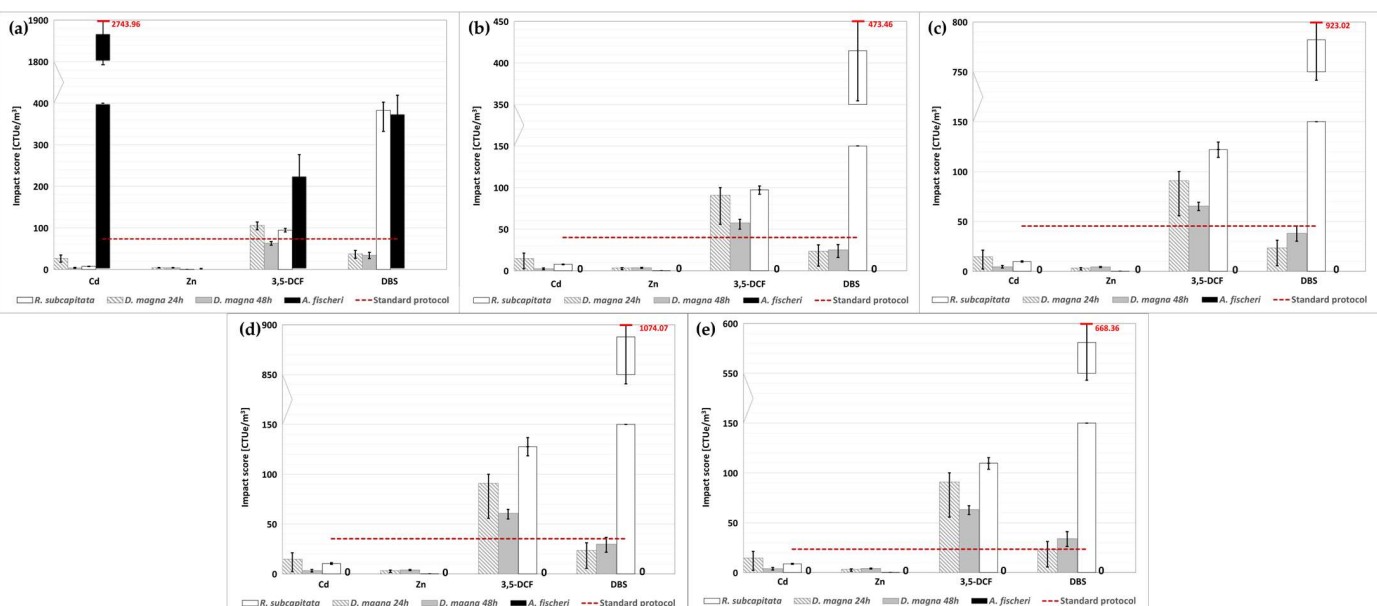

**Figure 3.** Environmental footprint related to freshwater ecotoxicity calculated with the alternative protocol. (**a**) WWTP A; (**b**) WWTP B, routine time; (**c**) WWTP B, harvest time; (**d**) WWTP C, CAS; (**e**) WWTP C, MBR (the red dotted line represents the conventional impact score calculated without considering the substances detected above the LOQ).

The difference that was found between the two diverse types of reference compounds (organics and metals) could be explained by the fact that the corresponding CFs were produced from bioassays that had a high sensitivity to metals. As a direct consequence of this, the application of CFs in organics is now a more viable option. Furthermore, biological studies on organisms from different levels of the trophic web suggest that effluents may contain a mixture of pollutants that are particularly effective against certain organisms. In contrast to the effluents of other plants, the discharged water from WWTP A is characterized by a mixture in which the synergistic action of the individual chemicals drives the luminescent bacteria response.

The impact score from the four baseline toxicity tests was then added together and represented in a box plot for each reference component. This overall representation has been developed for each wastewater treatment plant, as shown in the graphs (Figure 5).

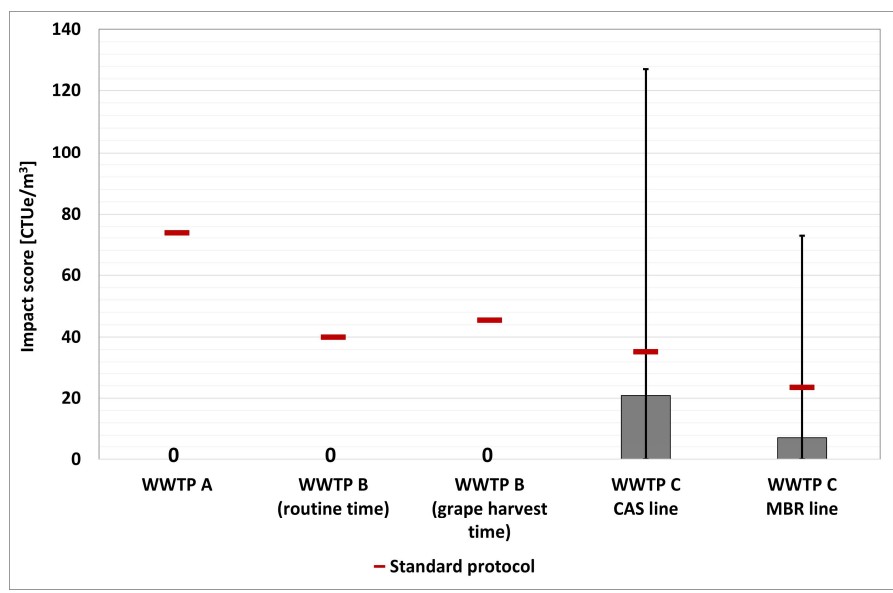

**Figure 4.** Environmental footprint related to the freshwater ecotoxicity calculated with the alterative protocol starting from the outcomes of the *A. cepa* test. "0" indicates that the impact score is nil. The red stroke represents the conventional impact score calculated without considering the substances detected above the LOQ.

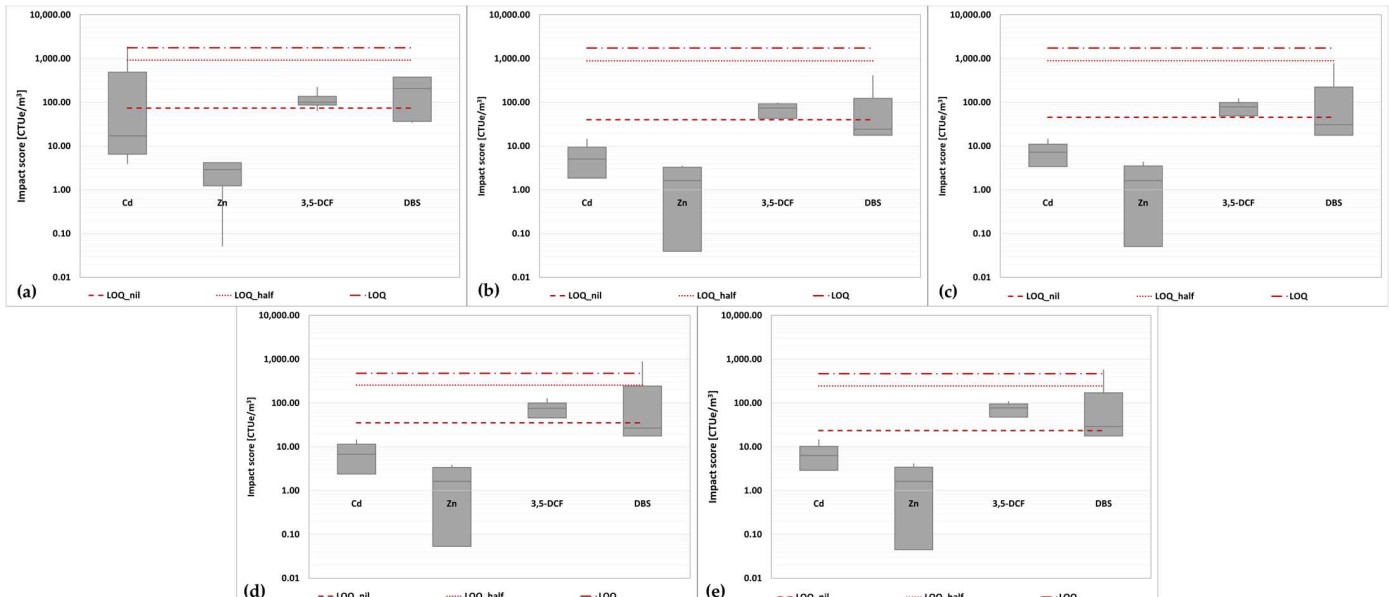

**Figure 5.** (**a**) WWTP A; (**b**) WWTP B, routine time; (**c**) WWTP B, harvest time; (**d**) WWTP C, CAS; (**e**) WWTP C, MBR: box plot of the impact scores obtained from the four baseline toxicity assays (*D. magna* 24 h, *D. magna* 48 h, *R. subcapitata* and *A. fischeri*). The three red dotted lines represent the conventional impact score calculated using the three ways mentioned above.

Because the estimation of the impact score is affected by a number of criticalities, the evaluation of the unique impact score for each WWTP would be characterized by a low degree of robustness. Instead, a more reliable comparison of the various options can be made. Based on the box plot representation, a thorough analysis reveals that WWTP A is the worst of the three.

### 3.2.2. Comparing the Protocols: Human Toxicity/Cancer: Outcomes

Based on the various bioassays performed, the innovative procedure developed four scenarios for each wastewater treatment plant in this impact category (see Figure 6). Due to the fact that just one substance was employed as a reference chemical for each bioassay, the summarizing depiction of the data is not feasible.

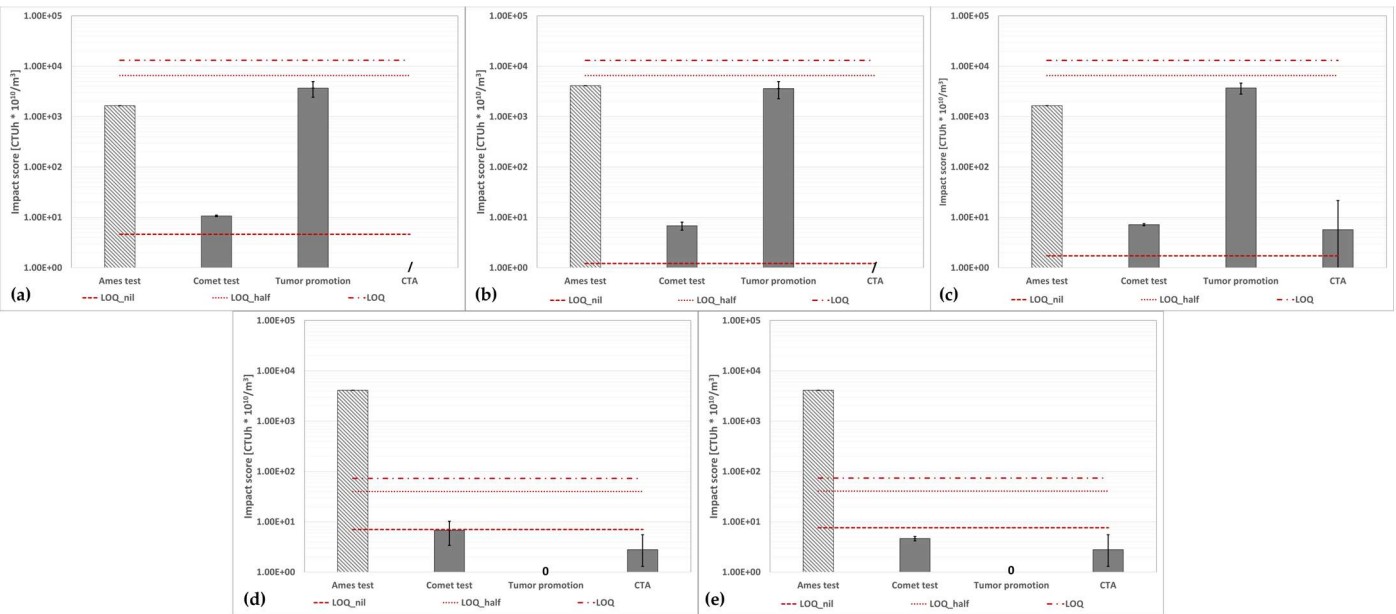

**Figure 6.** (**a**) WWTP A; (**b**) WWTP B, routine time; (**c**) WWTP B, harvest time; (**d**) WWTP C, CAS; (**e**) WWTP C, MBR: calculated environmental footprint using the alternative protocol for the human toxicity/cancer category (the three red dotted lines represent the conventional impact score calculated with the abovementioned three ways). The character "/" indicates that the assay was not performed. The Ames test histogram shows that the impact score is lower than the displayed value.

A thorough examination reveals that the environmental impact obtained by the Ames test is always greater than the impact score obtained by the Comet test. This situation is consistent with expectations because the Ames test looks for point reverse mutations (frameshift mutations and base-pair substitutions, depending on the strain) with and without metabolic activation (S9), whereas the Comet assay looks for damaged DNA in human leukocytes. Furthermore, a comparison of the other bioassays reveals that the tumor promotion test has a higher impact score than the CTA test footprint in all WWTPs (except WWTP C where the trend is inverted since the impact associated with the tumor promotion is nil). The graphs show that WWTP A and B (during both monitoring periods) have larger environmental footprints than WWTP C (in both treatment lines). A more detailed analysis reveals that WWTP A has a significant environmental footprint in comparison to WTP B, whereas the two treatment lines in WWTP C have a similar impact score.

The criticalities that have emerged from the proposed elaboration, which were carried out based on experimental data obtained on emissions considered as a whole, confirm the doubts and gaps that have yet to be filled and have already been highlighted by several authors. This was achievable because the proposed elaboration was carried out on the basis of experimental data obtained on emissions considered as a whole.

Some of the critical issues raised in our work have been investigated by various authors in relation to important production sectors. Among the others, building materials is one of the industries where the LCA technique causes challenges due to the presence of complex matrices. It is worth noting the observation of [52], who point out that the toxicity of mixtures (in this case, emissions) is frequently underestimated and, on the other hand, that in most cases where the LCA calculation of a product or process is carried out, more emphasis is placed on environmental aspects in line with individual country

policies, including, without a doubt, global warming. Any changes in the life cycle of the products and processes under consideration are thus primarily targeted at addressing the ever-increasing concentration of greenhouse gases. The same is true for the reduction of energy consumption. In this approach, the ecotoxicological effects are regarded as a type of collateral or secondary impact [53,54]. In conclusion of a reasoned critical review on the toxicity of building materials, ref. [52] state that LCA protocols, which are the optimal tool for calculating the various impacts, should be improved in terms of toxic effect estimation in order to at least consider the phenomena of bioaccumulation and the mixture effect. Emphasis on the same critical issues is also placed by [55], who propose the LCA on the Danish pork supply chain: not only is toxicity frequently overlooked, with most studies focusing on effects such as eutrophication and acidification, but exposure to multiple substances is also often overlooked when it is taken into account. LCA methods should consider this and not be confined to the additive approach. A further emblematic case is represented by the pharmaceutical industry: the estimation of the impact of production cycle emissions must necessarily consider a multiplicity of substances (active ingredients and their precursors/reaction by-products) absent from the LCA inventories. This leads to underestimated and, in any case, unreliable values [56]. A further and, in many ways, more serious consequence of the absence of chemicals in inventories is the possible implicit conclusion of their absence in the real scenario being studied. This is what [57] highlight with regard to another industrial sector of extreme importance from an economic and environmental point of view, such as the textile one. An additional perspective of improvement of LCA protocols is suggested by [58], who underline, in the exemplary case of titanium dioxide-based nanomaterials, the need to introduce EFs that better simulate the real environmental conditions (in this case, the formation of radical species due to the action of UV radiation).

The European Union, on the other hand, strongly encourages the use of tools such as OEF and PEF to comply with the UN Life Cycle Initiative project "Linking the UN Sustainable Development Goals to Life Cycle Impact Pathway Frameworks," which recommends the use of LCA to monitor SDGs at the corporate level. Similarly, the adoption of the OEF/PEF, due to the fact that it includes the categories of human toxicity and freshwater ecotoxicity, goes in the direction of compliance with EU policies and strategies, such as the CSS (Chemicals Strategy for Sustainability, SDG, 15: Toxicity-free environments), the CEAP (Circular Economy Action Plan), the F2F (Farm to Fork), and the BS (Biodiversity Strategy) [59]. Nonetheless, ref. [60] critically review a number of studies evaluating the impact of waste reutilization in agriculture in terms of toxicity: in many cases, toxicity can only be estimated by considering trace pollutants or, at the very least, a very small portion of organic substances. This is due to the fact that models and inventories only (necessarily) consider a limited group of substances, excluding abiotic and biotic degradation by-products, toxins, and particulate matter. In fact, other authors have attempted a similar approach to ours, calculating the environmental risk associated with the use of digestate as a soil conditioner and considering it as a single matrix [61]. They calculated the EF parameter (see Equation (2), Section 2.3) for terrestrial ecotoxicity, which will be used to calculate the CF value once the FF and XF values for exposure and fate in the ecosystem have been estimated.

## 4. Conclusions

This study was carried out in order to adapt a valuable, versatile, and extremely promising instrument such as OEF/PEF to a sector such as wastewater purification, which inherently fulfils an environmental task and may potentially be a hotspot for trace contaminants. What emerges once more from this study (compared to what has been presented in the scientific literature) is the limitation of the OEF/PEF approach where emissions, taken as complex matrices, are to be investigated.

In this case, our suggestion of employing the equivalent substances to be included in models for assessing possible consequences on human toxicity/non-cancer and freshwater

ecotoxicity might prove to be a workable technique in order to take into consideration effluents, emissions, or waste as a whole. In fact, the results of this study indicated that organisms with varied trophic roles and biological complexities (unicellular versus multicellular; prokaryotes versus eukaryotes) behave differently in terms of baseline toxicity (as expected). This shows that one cannot depend simply on a biological assay but rather must combine a battery of them, widening the range of quantifiable endpoints. Similarly, it is now evident that numerous reference substances ought to be chosen for use as toxicity equivalents. This would allow LCA systems to be utilized without the need to measure every chemical in the sample (which is impractical) and, more importantly, without the need to take into account the effects that the substances individually and collectively exert.

**Author Contributions:** Conceptualization, G.B., M.M. and R.P.; Methodology, G.B., M.M. and R.P.; Formal Analysis, G.B. and M.M.; Investigation, D.F., G.M., R.P., N.S., C.U. and I.Z.; Data Curation, all.; Writing—Original Draft Preparation, G.B., M.M. and R.P.; Writing—Review and Editing, G.B., M.M. and R.P.; Visualization, G.B.; Supervision, G.B.; Project Administration, G.B.; Funding Acquisition, G.B. All authors have read and agreed to the published version of the manuscript.

**Funding:** This research was funded by the University of Brescia ("Wat_Challenge Project" and "Smart_Wat Project" within the calls "Health&Wealth"; Principal Investigator: Giorgio Bertanza). Bando di Ateneo Health&Wealth2015 "The water challenge: smart models, tools and methods for assessing environmental suitability and effects of green technologies on human health WAT_CHALLENGE), Prot. 9019, 05.04.16. SMART_WAT (Smart drinking- and waste-water treatment strategies for the protection and exploitation of natural water reservoirs), Bando di Ateneo Health&Wealth2017, B + LabNet.

**Data Availability Statement:** The original contributions presented in this study are included in the article. Further inquiries can be directed to the corresponding author.

**Conflicts of Interest:** The authors declare no conflict of interest.

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
