# Peer review of "Beyond the Black Box of Life Cycle Assessment in Wastewater Treatment Plants: Which Help from Bioassays?"

_water, doi:10.3390/w15050960_

Round 1

Reviewer 1 Report

This is an interesting manuscript. Please see the following comments.

When answering te comments please provide me with a separate piece of paper that states the comments and then gives your answer and corrections as shown in the amended (non track changes version) of the revision together with the lines of the changed text, otherwise in most cases it is almost impossible to understand what changes were made and when. Thank you for your understanding

1) English language corrections are imperative. Please proofread throughout the document and correct. For example, in many cases you should cut the sentences in 2 sentences each and state in simple clear terms what is meant in each part

2) I understand that this is a paper written within the EU however it should not be inferred that all will use the JRC recommendations, so it should be written as eg "usually", or "in many cases" throughout.

3) I believe that ISO 14040 and ISO 14044, (ILCD) Handbook as well as Recommendation 2013/179/EU should be included in the list of references also

4) I believe that you shouldnt dwell in the introduction so much on general procedures and organizations since clearly you talk about WWTP which is very relevant for this kind of investigation

5) the last 3 paragraphs in lines 78-90 should be merged into one that states exactly what the design was and why. I do not understand what is meant by 

"The goal of this study was to critically examine the protocols provided for defining 83 risk to human health and the aquatic water ecosystem in order to understand their persis- 84 tent criticalities" I dont think that this was the aim of the study-what protocols you examined? you mean the strategy of measuring the chemicals only? what you mean by persistent criticalities? also I dont understand what you mean by  the calculation of OEF/PEF cannot yet officially 85 include toxicity categories, precisely because of the uncertainty in risk estimation, so how the risk to humans nd ecosystem is measured via the (ILCD) Handbook? if different endpoints are used in the two different approaches how can the approaches be then compared? please rewrite

6) Fig 1,2,3 and 4 should be merged in 1a, b, c and d

7) It is impossible to understand from the materials and methods the following

a) whether the subsets in Fig 5 were examined as part of this paper or if they are from other papers and they are used here. In the former situation you need to explain the protocols in more details

b) you state However, only the 131 direct emissions in the water bodies are considered in the evaluation, disregarding other 132 direct emissions and indirect emissions such as raw material consumption, the construc- 133 tion phase, energy consumption, sludge production, and waste production. This decision 134 is motivated by the strong correlation between the residual pollution in the discharged 135 water and the two impact categories chosen but also you state in the introduction about the the PEF/OEF approach. so here only one route was considered or more than one?

c) you state in great detail eg lines 201-205, lines 206-231, the background of the development of the indicators and this should be mentioned in the discussion

8) there are mistakes of haste eg paragraph in lines 192-196 is duplicated

9) you never really describe what you did exactly in the methods since 

you state A thorough examination of the standardized approach reveals some flaws in the 244 methodology that introduce uncertainty into the final impact score. The following are the 245 weaknesses

where is this examination and how it can be depicted in the results. it is unclear to me

10) the same about Fig 6  to 16(!) first of all it is not correct to include so many figures-some that are least commented should be moved to supplementary material. secondly what do these figures depict? the discrepancy between the 2 approaches? then why are they under 3.2. Alternative approach? also about tables 3 to 5 I am sorry but I cannot understand how they show that there is uncertainty, you do not really explain these results

11) maybe a good alternative would be to do a separate RESULTS section where you clearly explain what is shown in the tables and figures and then a discussion section where you explain the drawbacks through the comparison. In any case right now the plurality of Figures does not help the comprehension of the paper

12) regarding the importance and the evaluation of the comet assay in aquatic organisms please also quote if possible

Emmanouil, C.Smart, D.J.Hodges, N.J.Chipman, J.K. Marine Environmental Research2006,

Reviewer 2 Report

This paper exhbited good language and it meets the scope of your special issue. This study is novelty and creative and I suggest to accept this paper.

Reviewer 3 Report

In this manuscript, Menghini et al. described “Beyond the black box of life cycle assessment in the wastewater treatment plants: which help from the bioassays?” in detail. Samples are properly characterized, and the activities are excellent. However, at this stage there are still many problems and I therefore suggest a major review for this manuscript keeping in mind the following questions.

1) English language is poor and further revision is needed to improve the revised manuscript.

2) The author mentioned that “Since it is impossible to measure every pollutant (primary and secondary) present in an emission” the term impossible does not seem to be good here, rather difficult, or hard will be ok.

3) No correlation is present between different paragraphs of the introduction, please correlate these to attract more reviewers.

4) the sentence “were conducted to account for both routine time and grape harvest period. [40] provides more” has a problem as full stop before [40] and then provides are not correct.

5) A definite relationship must exist between the different portions of the manuscript, for example abstract and conclusion are written in past tense, experimental part is written in past, result and discussion are written in present tense, while in introduction, shortcomings are written in present tense, review literature is written in past tense. No correlation is seen in the manuscript. As you mention that “The environmental footprint of the aforementioned WWTPs is calculated using only two impact categories:” the word “is” not proper but “was” is acceptable.

6) Some very important citations are missing.

i) R. Nazir, M. Khan, R. U. Rehman, S. Shujah, M. Khan, M. Ullah, A. Zada, N. Mahmood, I. Ahmad, Adsorption of selected azo dyes from an aqueous solution by activated carbon derived from Monotheca buxifolia waste seeds, Soil Water Res. 15 (2020) 166-172.

ii) M. Ullah, R. Nazir, M. Khan, W. Khan, M. Shah, S. G. Afridi, A. Zada, The effective removal of heavy metals from water by activated carbon adsorbents of Albizia lebbeck and Melia azedarach seed shells, Soil Water Res. 15 (2020) 30-37.

7) Too many abbreviations are used in the manuscript, try to minimize them.

8) The conclusion is too long and not precise, the respected authors are requested to write a precise conclusion.

Reviewer 4 Report

The authors investigated wastewater treatment plants for life cycle assessment to safeguard public health. The appropriate studies are always welcome in human needs. Some important points are important to be addressed before going to consider the possible publication in this journal.

-The English language needs to check carefully in the revision stage because of many careless mistakes in many positions.

-The Figures quality needs to be improved in the revision stage.

-References: There are many references that are not adjacent to this study. The authors need to take notes in the revision stage and cite relevant references including high-impact journals as defined by the Awual group to make the manuscript to a broad range of readers.

-Abstract: This section needs to be improved by presenting novel findings of the experimental outcomes in the revision stage.

-Introduction: Novel materials are growing attention for diverse uses as reported by the Awual group according to ScienceDirect. The authors need to indicate such points for a broad range of readers. Moreover, the authors need to cite high-impact articles to make the manuscript high-level. The following specific articles may take be noted in the revision stage of Journal of Molecular Liquids, 371 (2023) 121125; Microchemical Journal 162 (2021) 105868; Journal of Molecular Structure, 1276 (2021) 134795.

-Scientists considered the cost-effectiveness of each study. The authors need to indicate such a point in the revised manuscript.

-Conclusion also needs to be rewritten. Include the following: new concepts and innovations demonstrated in this study, a summary of findings, a comparison with findings by other workers, and a concluding remark.

I would like to see the revised manuscript.